# Vitamin D Status and Reference Intervals Measured by Liquid Chromatography–Tandem Mass Spectrometry for the Early Adulthood to Geriatric Ages in a South Korean Population during 2017–2022

**DOI:** 10.3390/nu16050604

**Published:** 2024-02-22

**Authors:** Jooheon Park, Yongjun Choi, Seon Cho, Hyeran Park, Suyoung Kim, Han-Ik Cho, Eun-Hee Nah

**Affiliations:** 1Department of Laboratory Medicine, Chonnam National University Hwasun Hospital, Hwasun 58128, Republic of Korea; goodjusa@naver.com (J.P.); azarsis@hanmail.net (Y.C.); 2Health Promotion Research Institute, Korea Association of Health Promotion, Seoul 07572, Republic of Korea; dduddi3755@hanmail.net (S.C.); hyeran826@naver.com (H.P.); sy.kahp@gmail.com (S.K.); 3MEDIcheck LAB, Korea Association of Health Promotion, Seoul 07572, Republic of Korea; hanik@snu.ac.kr

**Keywords:** 25-hydroxyvitamin D, liquid chromatography–tandem mass spectrometry, 25(OH)D deficiency

## Abstract

This study aimed to describe the latest 25-hydroxyvitamin D (25(OH)D) status of the South Korean population aged ≥ 20 years using 25(OH)D concentrations measured by liquid chromatography–tandem mass spectrometry and to determine the factors associated with total 25(OH)D concentrations. This cross-sectional, retrospective study consecutively selected 119,335 subjects with a median age of 57 (20–101) years who underwent health checkups among 13 Korean cities during 2017–2022. The total 25(OH)D concentration was 54.5 ± 24.0 nmol/L (mean ± SD). The 7.6%, 47.5%, and 82.9% of participants had 25(OH)D less than 25, 50, and 75 nmol/L, respectively. The prevalence of 25(OH)D deficiency (<25 nmol/L) was higher in females than in males (8.9% vs. 6.1%) and varied between age groups, decreasing in older subjects. Those aged 20–29 years had the highest prevalence of 25(OH)D deficiency (23.0% in females and 20.1% in males), which also varied between cities. In the adjusted model, female sex, older age, summer and autumn seasons, lower body mass index (<25 kg/m^2^), and lower high-sensitivity C-reactive protein concentration (<1 mg/L) were associated with higher total 25(OH)D concentrations. This study could provide an exact understanding of the status of vitamin D and help devise strategies to prevent vitamin D deficiency among the Korean population.

## 1. Introduction

Vitamin D deficiency is considered a common worldwide health problem [1]. Adequate vitamin D status plays an essential role in skeletal bone health. A low vitamin D status is associated with muscle weakness, fragility fractures, bone loss, and falls in older people [2,3]. Besides its role in musculoskeletal health, a low vitamin D status is also correlated with non-skeletal disorders including cancer, diabetes, and cardiovascular and autoimmune diseases [4,5,6,7], which are very important public health issues. Estimations of vitamin D status are key to establish public health strategies. Some studies have found that the prevalence of vitamin D deficiency differs with population characteristics such as residence region and ethnicity [8,9,10]. The vitamin D status should be estimated using a nationally representative population. Moreover, it is also important to establish the reference intervals (RIs) that describe the ranges found in a healthy population as RIs of vitamin D may vary with demographic background.

Serum 25-hydroxyvitamin D (25[OH]D) is the best indicator for measuring vitamin D levels, as it reflects both the vitamin D from food and supplements and the vitamin D that the body makes from cholecalciferol (vitamin D_3_) when exposed to the sun’s UV–B rays on 7-dehydrocholesterol [11,12]. In 2021, the US Preventive Services Task Force issued a new recommendation, stating that there is not enough evidence to evaluate the pros and cons of screening for vitamin D deficiency in adults without symptoms [13]. The recommendation mentioned that although liquid chromatography–tandem mass spectrometry (LC-MS/MS) is considered the reference assay, it is a complicated process that is liable to variations and errors, including interference from other chemical compounds. However, LC-MS/MS is currently considered the gold standard for measuring vitamin D metabolites and has become more accessible for routine and high-volume analyses [14,15]. The Vitamin D Standardization Program has also developed a reference measurement system to establish international standardization of 25(OH)D measurements [16].

While the status of vitamin D has been previously described in some populations [17,18,19], these studies had relatively limited populations visiting hospitals or unstandardized assay methods such as immunoassay or radioimmunoassay-based vitamin D measurements. Therefore, current information on the status of vitamin D is needed based on 25(OH)D levels measured by LC-MS/MS in the general Korean population of all adult ages. This study aimed to describe the latest 25(OH)D status of the South Korean general population aged ≥ 20 years during 2017–2022 using 25(OH)D data measured by LC-MS/MS and to identify the associated factors of total 25(OH)D concentrations.

## 2. Materials and Methods

### 2.1. Study Population

This is a cross-sectional retrospective study. The study subjects were consecutively selected from examinees who underwent health checkups that included the assessment of 25(OH)D concentration at 17 health-promotion centers in South Korea from January 2017 to December 2022. This study analyzed 119,335 of the 123,124 eligible subjects (Figure 1). These 17 health-promotion centers are affiliated with the Korea Association of Health Promotion. The National Health Insurance System (NHIS) of South Korea provides medical examinations every two years for its entire population. The 17 health-promotion centers included in this study perform around 10% of the health checkups offered by the NHIS. All participants provided their medical history, and their subjective symptoms and signs were assessed during the health checkups. Their medical records were also reviewed. The participants were excluded if they had a history of cancer, myocardial infarction, or stroke. We defined vitamin D deficiency as <25 nmol/L, vitamin D insufficiency as 25 to 50 nmol/L, and sufficiency as >50 nmol/L by 25(OH)D level, according to the guidelines of the Endocrine Society [20].

The study protocol was reviewed and approved by the Institutional Review Board of the Korea Association of Health Promotion on 19 April 2023 (approval no. 130750-202304-HR-002). The need to obtain informed consent was waived since the study analyzed de-identified data that were obtained during previous health checkups.

### 2.2. Laboratory Measurements of Serum 25(OH)D

The health checkups included the drawing of venous blood after fasting overnight to measure vitamin D. Venous blood was collected in serum tubes with a clot activator. As per routine procedure, blood samples were centrifuged at 1500 rpm for 10 min when they reached the lab and kept at 4 °C until they were analyzed within 7 days in the core laboratory. The serum samples were subjected to derivatization, hexane extraction, and trypsin digestion, followed by analysis with the ACQUITY Ultra-Performance Liquid Chromatography System and Xevo TQ-S Mass Spectrometer (Waters, Milford, MA, USA) to measure the concentration of the 25(OH)D_2_ and 25(OH)D_3_. The samples were treated with zinc sulfate to precipitate proteins and then spiked with an internal standard (d6 25[OH]D_2_ and d6 25[OH]D_3_) that was eluted in methanol. The samples were then subjected to online solid-phase extraction (RECIPE, Munich, Germany). The method was calibrated using ClinCal Serum Calibrators (RECIPE) and enabled the equimolar quantification of the two predominant 25(OH)D species: 25(OH)D_3_ and 25(OH)D_2_. The LC-MS/MS method was standardized and traceable to the National Institute of Standards and Technology (Gaithersburg, MD, USA). The limit of detections of 25(OH)D_2_ and 25(OH)D_3_ were 2.5 nmol/L and 0.75 nmol/L, respectively, with respective imprecision CVs of 7% and 6%; the total was <10%. The accuracy of the measurement of 25(OH)D_2_ and 25(OH)D_3_ was between 96.5% and 108% of standard materials. Our assay accuracy is verified by our regular participation in external quality assurance programs, such as the College of American Pathologists CAP Proficiency Testing/Quality Management program and the Vitamin D External Quality Assessment Scheme DEQAS for vitamin D. The calculation of the total serum 25(OH)D concentration involved adding 25(OH)D_2_ and 25(OH)D_3_ together.

### 2.3. Statistical Analysis and Calculation of Reference Intervals for Serum 25(OH)D_2_, 25(OH)D_3_, and Total 25(OH)D

Statistical analyses were performed using SAS software (version 9.4, SAS Institute, Cary, NC, USA). Data are presented as mean ± SD or frequency (percentage) values. Differences in total 25(OH)D, 25(OH)D_2_, and 25(OH)D_3_ among age groups and months of blood collection were analyzed using one-way ANOVA with Bonferroni’s post hoc comparison. The differences among sex, body mass index (BMI) groups, and high-sensitivity C-reactive protein (hs-CRP) were analyzed using *t*-tests. The levels of 25(OH)D_2_, 25(OH)D_3_ and the total 25(OH)D were analyzed for calculating RIs according to guideline C28-A3 of the Clinical and Laboratory Standard Institute [21]. A nonparametric method was used to determine the RIs for the total 25(OH)D, 25(OH) D_2_, and 25(OH)D_3_ concentrations (the 2.5th and 97.5th percentiles). The chi-squared test was used to compare the prevalence rates of total 25(OH)D concentration ranges of <25, <50, <75, and ≥75 nmol/L according to age and sex. The factors related to the total 25(OH)D concentration were identified by performing multiple linear regression analyses, after adjusting for sex, age, month of blood collection, BMI, and hs-CRP. The Durbin–Watson statistic had a value of 1.951, which indicated no autocorrelation in the sample. Two-sided *p* < 0.05 was considered to indicate statistical significance.

## 3. Results

### 3.1. Characteristics of Study Subjects

This study included 119,335 subjects, 58,151 (48.7%) males and 61,184 (51.3%) females. The age of the study subjects was 55.4 ± 11.3 years (median: 57 years; range: 20–101 years). The age distribution was as follows: 1981 subjects (1.6%) aged < 20–29 years, 9286 (7.8%) aged 30–39 years, 22,175 (18.6%) aged 40–49 years, 38,049 (31.9%) aged 50–59 years, 37,557 (31.5%) aged 60–69 years, and 10,287 (8.6%) aged ≥ 70 years. The BMI was 24.3 ± 3.4 kgm^2^.

### 3.2. Distributions of Total 25(OH)D, 25(OH)D_2_, and 25(OH)D_3_ Concentrations by Sex, Age, Season, and Demographic Characteristics

The total 25(OH)D concentration in the study subjects was 54.5 ± 24.0 nmol/L. The mean 25(OH)D concentration was significantly lower in males than in females (52.5 vs. 56.3 nmol/L, *p* < 0.001). The total 25(OH)D concentration increased significantly with age (*p* < 0.001) (Table 1, Appendix A), and was lower in subjects with obesity (BMI ≥ 25 kg/m^2^) (51.8 vs. 56.0 nmol/L, *p* < 0.001) and a high hs-CRP concentration (≥1 mg/L) (52.3 vs. 54.5 nmol/L, *p* < 0.001) (Table 1). The relationship between the mean total 25(OH)D concentration and month of blood collection revealed a seasonal variation. The mean total 25(OH)D concentration peaked in the summer (June–August) and was lowest in the winter (December–February) (57.5 vs. 50.0 nmol/L, *p* < 0.001) (Table 1, Appendix A). These distribution trends were similar for 25(OH)D_2_ and 25(OH)D_3_.

### 3.3. RIs for Serum 25(OH) D_2_, 25(OH)D_3_, and Total 25(OH)D

Table 2 lists the RIs (95% confidential intervals) for serum 25(OH)D_2_, 25(OH)D_3_, and total 25(OH)D. The RI for serum 25(OH)D among all study subjects was 19.3–110.3 nmol/L. The range of total 25(OH)D RIs was wider in females than in males. The upper and lower limits increased with age except for the lower limit among those older than 70 years (Table 2). The 25(OH)D_2_ was detected in 9.2% of the study subjects, and the detection frequency varied with sex and age, being higher in females than in males (10.3% vs. 8.0%) and older subjects (*p* < 0.01) (Appendix A).

### 3.4. Prevalence of 25(OH)D Deficiency for Different Cutoff Values

The 7.6%, 47.5%, and 82.9% of participants had 25(OH)D less than 25, 50, and 75 nmol/L, respectively. The 25(OH)D concentration was ≥75 nmol/L in 17.2% (12.9% in males and 21.2% in females). The prevalence of 25(OH)D deficiency (<25 nmol/L) was higher in females than in males (8.9% vs. 6.1%) and also varied with age, which was lower in older subjects (*p* < 0.001). Those aged 20–29 years had the highest prevalence of 25(OH)D deficiency (<25 nmol/L) (23.0% in females and 20.1% in males) (Table 3). The prevalence of 25(OH)D deficiency also varied between cities, being low in Jeju and Changwon and high in Chungju, Incheon, and Seoul (Appendix A).

### 3.5. Factors Associated with Total Serum 25(OH)D Concentration

In the adjusted model, female sex, older age, summer and autumn seasons, lower BMI (<25 kg/m^2^), and lower hs-CRP concentration (<1 mg/L) were associated with higher total 25(OH)D concentration (*p* < 0.001) (Table 4).

## 4. Discussion

The mean total 25(OH)D concentration by LC-MS/MS method in the present study was 54.5 nmol/L for the South Korean general population, which is close to the cutoff of the sufficiency level (>50 nmol/L). The 25(OH)D concentration was much higher for the factors of female sex, older age, summer and autumn seasons, lower BMI, and lower hs-CRP concentration. Moreover, the prevalence of vitamin D deficiency (<25 nmol/L) was 7.6%, which was not much higher than that in other developed countries. However, it was high in young adults, especially in young females aged <30 years. This study had the following strengths: (1) A large number of subjects (*N* = 119,335) with a median age of 57 (range: 20–101) were enrolled from the general population to analyze the status and reference intervals of vitamin D; (2) it analyzed nationally representative data on vitamin D and its components for the South Korean population; and (3) use of the LC-MS/MS, which is a standard method for measuring the vitamin D concentration in health checkups.

Although the vitamin D mega-trials have shown a lack of evidence for beneficial effects against hard-disease outcomes, such as cardiovascular disease, cancer, fracture, or fall, the evaluation of vitamin D status is still necessary to prevent vitamin D deficiency in the population because it has shown possible beneficial effects on arterial function, bone mineral density of the hips and spine, and lung function, especially in individuals with a vitamin D deficiency [2,3,4,22,23].

The mean 25(OH)D concentration in South Korea was close to the cutoff of the sufficiency level (>50 nmol/L). However, it varied by age, with a tendency to be below 50 nmol/L in those younger than 50 years. The US National Academy of Medicine suggests a 25(OH)D concentration above 50 nmol/L as a measure of ‘sufficiency’ [24]. A concentration of <50 nmol/L does not imply a definitive diagnosis of vitamin D deficiency, nor does it indicate a need for supplementation intervention. However, screening for vitamin D deficiency should be based on that criterion, especially for those who have certain diseases such as hepatic failure, chronic kidney disease, osteoporosis or obesity, or who belong to specific racial subgroups [25]. Our study found that 39.9% of the subjects had ‘inadequate’ concentrations (25–50 nmol/L), which were much higher in males aged < 60 years and females aged < 50 years. Moreover, the mean 25(OH)D concentration was higher in females. There is controversy about sex differences in 25(OH)D concentrations among studies [8,18,26,27]. A USA population study [8] found that 25(OH)D concentration was higher in non-Hispanic white females; on the other hand, other studies found higher levels in males [18,26], or no sex differences [27]. These discrepancies might be caused by lifestyle differences such as engagement in outdoor activities, supplement use, and sunscreen use; however, we could not ascertain this since lifestyle was not assessed in this study.

Most of the expert group recommended that a serum 25(OH)D level of 25 nmol/L be regarded as the minimum threshold for vitamin D status and/or a marker of vitamin D deficiency risk. This cutoff value was set based on preventing nutritional rickets and osteomalacia [28,29,30]. The overall prevalence of vitamin D deficiency (<25 nmol/L) was 7.6% in the present study, which was similar to the prevalence rates (range: 5.9–13%) observed in other studies [8,9,10] based on data from the standardized LC-MS/MS method. We found that the prevalence of vitamin D deficiency was lower in South Korean adults than in reports from other Asian countries [31,32]. However, this prevalence of vitamin D deficiency (<25 nmol/L) varied with age and sex in our study, being higher in females and young adults aged < 30 years, and the highest in young females aged < 30 years. Strategies for vitamin D deficiency prevention should be considered for this subgroup.

In addition to the mean 25(OH)D concentration varying by age and sex, it also varied depending on the month of blood collection. June to August is the summer season in Korea as Korea is located in the middle latitude of the Northern Hemisphere. The mean total 25(OH)D concentration peaked in the summer months (June–August) and was lowest in winter (December–February). This trend was also present for 25(OH)D_3_ but not for 25(OH)D_2_. The 25(OH)D_2_ concentration varied little with the month of blood collection, which is consistent with a previous study [33] finding that the 25(OH)D_2_ concentration varied little throughout the year. However, the 25(OH)D_2_ concentration varied with age and was higher in those older than 50 years in our study. The presence of 25(OH)D_2_ was considered a result of the prescription of high-dose ergocalciferol [8]. According to Cashman et al. [34], age and vitamin D supplementation positively influenced 25(OH)D_2_ levels. Furthermore, there were other factors associated with 25(OH)D concentration in our study, such as obesity and inflammation, which were similar to the results of another previous study [8].

The prevalence of vitamin D deficiency also varied with the city of residence, being higher in large industrialized cities such as Seoul and Incheon. These cities have high proportions of young indoor office workers and shift workers. Some systematic reviews [35,36] stated that vitamin D deficiency is prevalent among shift and office workers. However, we could not confirm this as the reason in the present study since the occupations of study subjects could not be obtained.

Our study had some limitations. First, we could not obtain the vitamin D intake data such as fortified foods and supplements. Second, information about lifestyles such as engagement in outdoor activity, occupation, and sunscreen use, which are important confounders for assessing vitamin D status, could not be obtained in this study. Lastly, this study had a cross-sectional design, so the causal relationship between the factors that have statistical significance and the level of vitamin D could not be determined.

## 5. Conclusions

Although the mean 25(OH) concentration in the South Korean population was close to the ‘sufficient’ level of 50 nmol/L, half of the population did not reach this level, mostly comprising males aged < 60 years and females aged < 50 years. Moreover, the prevalence of vitamin D deficiency in the South Korean population was not much higher than that in other developed countries. However, the deficiency was high in young adults, especially in young females aged < 30 years. This study could provide an exact understanding of the status of vitamin D and help devise strategies to prevent vitamin D deficiency among the Korean population.

## Figures and Tables

**Figure 1 nutrients-16-00604-f001:**
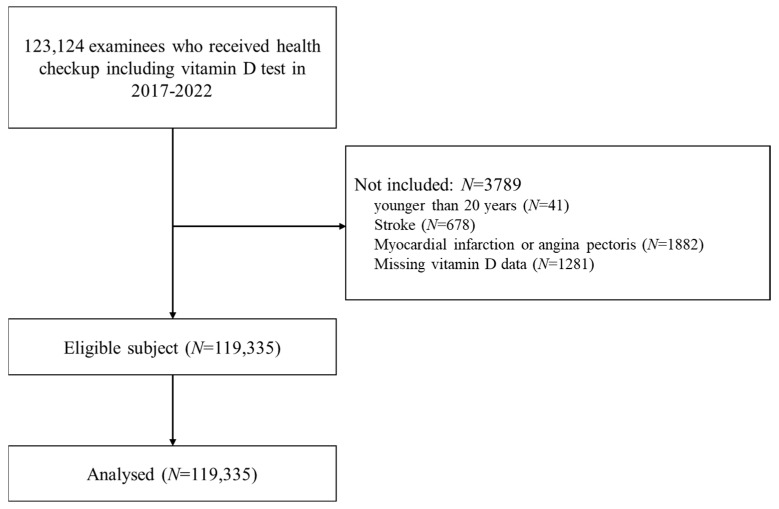
Flow chart of the study.

**Table 1 nutrients-16-00604-t001:** Measured 25(OH)D_2_, 25(OH)D_3_, and 25(OH)D concentrations.

		N	25(OH)D_2_	*p*-Value	Multiple Comparison	25(OH)D_3_	*p*-Value	Multiple Comparison	Total 25(OH)D	*p*-Value	Multiple Comparison
			Mean ± SD (nmol/L)			Mean ± SD (nmol/L)			Mean ± SD (nmol/L)		
All		119,335	0.5 ± 2.53	-		54.0 ± 24.0	-		54.5 ± 24.0	-	
Sex											
	Male	58,151	0.43 ± 2.2	<0.001		52.3 ± 21.0	<0.001		52.5 ± 21.0	<0.001	
	Female	61,184	0.6 ± 2.8			55.5 ± 26.5			56.3 ± 26.5		
Age, year											
	20–29 ^a^	1981	0.33 ± 2.25	<0.001	a,b,c < d < e < f	39.8 ± 19.0	<0.001	a < b < c < d < e < f	40.0 ± 19.0	<0.001	a < b < c < d < e < f
	30–39 ^b^	9286	0.38 ± 2.18			46.0 ± 20.8			46.3 ± 20.8		
	40–49 ^c^	22,175	0.38 ± 2.08			48.5 ± 21.0			48.8 ± 21.0		
	50–59 ^d^	38,049	0.45 ± 2.15			54.3 ± 23.3			54.8 ± 23.3		
	60–69 ^e^	37,557	0.63 ± 2.85			58.0 ± 25.0			58.5 ± 25.0		
	≥70 ^f^	10,287	0.75 ± 3.63			60.0 ± 27.0			60.5 ± 27.0		
Month											
	December–February ^a^	30,839	0.5 ± 2.25	<0.001	a,c,d < b	49.5 ± 24.3	<0.001	a < b < d < c	50.0 ± 24.3	<0.001	a < b < d < c
	March–May ^b^	26,608	0.75 ± 2.75			52.5 ± 25.3			53.3 ± 25.3		
	June–August ^c^	31,239	0.5 ± 2.5			57.0 ± 22.8			57.5 ± 22.8		
	September–November ^d^	30,649	0.5 ± 2.5			56.3 ± 23.3			56.8 ± 23.5		
BMI, kg/m^2^											
	<25	71,253	0.53 ± 2.65	<0.001		55.5 ± 25.5	<0.001		56.0 ± 25.5	<0.001	
	≥25	45,353	0.45 ± 2.33			51.3 ± 21.3			51.8 ± 21.3		
hs-CRP, mg/L											
	<1	107,790	0.5 ± 2.55	0.625		54.0 ± 23.8	<0.001		54.5 ± 23.8	<0.001	
	≥1	2183	0.48 ± 2.5			51.8 ± 22.3			52.3 ± 22.3		

^a,b,c,d,e,f^: Different letters indicate a significant difference between groups using one-way ANOVA with post hoc comparison test. 25(OH)D, 25-hydroxyvitamin D; BMI, body mass index; hs-CRP, high-sensitivity C-reactive protein.

**Table 2 nutrients-16-00604-t002:** Reference intervals (95% confidence intervals) for serum 25(OH)D_2_, 25(OH)D_3_, and 25(OH)D by sex and age.

		25(OH)D_2_, nmol/L				25(OH)D_3_, nmol/L				25(OH)D, nmol/L			
		2.5th	CI	97.5th	CI	2.5th	CI	97.5th	CI	2.5th	CI	97.5th	CI
Total		<LOD NA		5.0	(4.8, 5.0)	19.0	(18.8, 19.0)	109.8	(109.3, 110.5)	19.3	(19.0, 19.3)	110.3	(109.5, 111.0)
Sex													
	Male	<LOD NA		4.3	(4.0, 4.3)	20.3	(20.0, 20.5)	99.5	(98.8, 100.5)	20.5	(20.3, 20.8)	100.0	(99.0, 100.8)
	Female	<LOD NA		5.8	(5.5, 6.0)	17.8	(17.8, 18)	116.8	(115.8, 118.0)	18.0	(18.0, 18.3)	117.3	(116.3, 118.5)
Age, years													
	20–29	<LOD NA		3.5	(3.3, 4.5)	14.3	(13.5, 14.8)	86.3	(83.0, 92.3)	14.5	(13.8, 15.3)	88.0	(83.0, 91.3)
	30–39	<LOD NA		4.0	(3.8, 4.3)	16.3	(16.0, 16.8)	94.5	(92.8, 97.3)	16.5	(16.3, 16.8)	94.8	(93.3, 97.5)
	40–49	<LOD NA		4.0	(3.8, 4.3)	18.0	(17.8, 18.3)	95.3	(94.0, 96.5)	18.0	(17.8, 18.5)	95.5	(94.5, 96.8)
	50–59	<LOD NA		4.8	(4.5, 4.8)	19.8	(19.5, 20.0)	107.8	(106.8, 109.3)	20.0	(19.8, 20.3)	108.3	(107.0, 109.8)
	60–69	<LOD NA		5.8	(5.5, 6.0)	20.5	(20.3, 20.8)	116.0	(114.8, 117.3)	20.8	(20.5, 21.3)	116.8	(115.3, 117.8)
	Over 70	<LOD NA		6.8	(6.0, 7.8)	19.0	(18.3, 19.5)	122.3	(120.0, 125.3)	19.5	(19.0, 20.0)	123.0	(120.5, 126.0)

CI, 95% confidence interval; 25(OH)D, 25-hydroxyvitamin D; LOD, limit of detection; NA, not applicable.

**Table 3 nutrients-16-00604-t003:** Prevalence rates of serum total 25(OH)D deficiency, insufficiency, and sufficiency by age and sex.

		<25 nmol/L		25–49.9 nmol/L		50–74.9 nmol/L		≥75 nmol/L		* *p*
		*N*	(%)	*N*	(%)	*N*	(%)	*N*	(%)	
All subjects		9014	(7.6)	47,598	(39.9)	42,255	(35.4)	20,468	(17.2)	-
Males (age, years)										
	20–29	173	(20.1)	454	(52.7)	187	(21.7)	47	(5.5)	<0.001
	30–39	569	(11.1)	2630	(51.4)	1526	(29.8)	396	(7.7)	
	40–49	917	(7.4)	6035	(48.4)	4349	(34.9)	1157	(9.3)	
	50–59	910	(5.1)	7685	(43.4)	6895	(38.9)	2233	(12.6)	
	60–69	749	(4.3)	6720	(38.7)	7127	(41.1)	2761	(15.9)	
	Over 70	232	(5.0)	1688	(36.5)	1808	(39.0)	903	(19.5)	
	Total	3550	(6.1)	25,212	(43.4)	21,892	(37.7)	7497	(12.9)	
Females (age, years)										
	20–29	258	(23.0)	602	(53.8)	201	(18.0)	59	(5.3)	<0.001
	30–39	662	(15.9)	1929	(46.3)	1183	(28.4)	391	(9.4)	
	40–49	1312	(13.5)	4392	(45.2)	2876	(29.6)	1137	(11.7)	
	50–59	1581	(7.8)	7553	(37.2)	7017	(34.5)	4175	(20.5)	
	60–69	1250	(6.2)	6348	(31.4)	7141	(35.4)	5461	(27.0)	
	Over 70	401	(7.1)	1562	(27.6)	1945	(34.4)	1748	(30.9)	
	Total	5464	(8.9)	22,386	(36.6)	20,363	(33.3)	12,971	(21.2)	

* *p* in the chi-squared test. 25(OH)D, 25-hydroxyvitamin D.

**Table 4 nutrients-16-00604-t004:** Results from multiple linear regression analyses of serum 25(OH)D concentrations.

		Unadjusted Model			Adjusted Model		
		Coeff.	(95% CI)	*p*-Value	Coeff.	(95% CI)	*p*-Value
Sex, ref: female							
	Male	−1.45	(−1.6, −1.3)	<0.001	−0.88	(−1, −0.8)	<0.001
Age, year		0.17	(0.1, 0.2)	<0.001	0.17	(0.1, 0.2)	<0.001
Month, ref: December–February							
	March–May	1.22	(1.1, 1.4)	<0.001	0.94	(0.8, 1.1)	<0.001
	June–August	2.91	(2.8, 3.1)	<0.001	2.96	(2.8, 3.1)	<0.001
	September–November	2.66	(2.5, 2.8)	<0.001	2.89	(2.7, 3)	<0.001
BMI, ref: <25 kg/m^2^							
	≥25 kg/m^2^	−1.68	(−1.8, −1.6)	<0.001	−1.42	(−1.5, −1.3)	<0.001
hs-CRP, ref: <1 mg/L							
	≥1 mg/L	−0.87	(−1.3, −0.5)	<0.001	−0.74	(−1.1, −0.3)	<0.001

Adjusted *R*^2^ for adjusted model: 6.37%. Durbin–Watson D statistic for adjusted model: 1.951. 25(OH)D, 25-hydroxyvitamin D; BMI, body mass index; hs-CRP, high-sensitivity C-reactive protein; ref, reference; Coeff., coefficient.

## Data Availability

Data are contained within the article and Appendix A.

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
