# Peer review of "Vitamin D Status and Reference Intervals Measured by Liquid Chromatography–Tandem Mass Spectrometry for the Early Adulthood to Geriatric Ages in a South Korean Population during 2017–2022"

_nutrients, 2024, doi:10.3390/nu16050604_

Round 1
Reviewer 1 Report
Comments and Suggestions for Authors
I would like to thank to the editor the opportunity of reviewing this study, but the study presents many shortcomings in the analysis and results that seem incongruent. Could you answer the following questions:
What do you mean by RI (reference interval)?
What non-parametric method was used to determine the RIs and why?
What post hoc analysis was performed? justified
There is a lot of difference between the 2.5th and 97.5th percentile and the 95% CIs, this is rare. For what is this?
The author say: The 25(OH)D2 was detected in 9.2% of the study subjects, and the detection frequency varied with sex and age, being higher in females than in males (10.3% vs 8.0%) 166 and older subjects (P<0.01) ( data not shown).
Why don't they show the data?
I think the graphics should be integrated into the text, I think it does not exceed the number of pages.
Author Response
1. What do you mean by RI (reference interval)?
Answer)
In medicine and health-related fields, a reference interval (RI) is the interval of values that is deemed normal for laboratory markers. The RI for a given test is based on the results that are seen in 95% of the healthy population. Authors described the significance of RI in Introduction section (Page 2, Line 42-44) as follows: Moreover, it is also important to establish the reference interval (RI) that describes ranges found in a healthy population as RI of vitamin D may vary with demographic background.
2. What non-parametric method was used to determine the RIs and why?
Answer)
The International Federation of Clinical Chemistry (IFCC) recommendation focuses on the nonparametric procedure, and the Clinical and Laboratory Standards Institute (CLSI) guideline on reference interval estimation deals exclusively with the nonparametric approach. RIs have been widely calculated by estimating the 0.025 and 0.975 percentiles, 95% CI, from a healthy population. Direct sampling is a traditional approach that refers to generating RIs from preselected population with measurement and determination in order. However, direct sampling has the limitation in numbers of subjects. On the contrary, indirect sampling selects results from routine pathology results in a population. For overcoming the limitation in numbers, analytical approaches based on high- volume data have been conducted in clinical laboratory medicine. Appropriate statistical technique, such as the truncated maximum likelihood method, are then applied to determine RIs.
3. What post hoc analysis was performed? justified
Answer)
Authors performed an ANOVA with Bonferroni’s post-hoc test. Authors described this in Materials and Methods section (Page 3, Line 120-121).
4. There is a lot of difference between the 2.5th and 97.5th percentile and the 95% CIs, this is rare. For what is this?
Answer)
The 95% CI means confidential intervals which is expected to contain the 2.5th or 97.5th value being estimated. Authors checked and rearranged the Table 2 in order not to confuse readers.
5. The author say: The 25(OH)D2 was detected in 9.2% of the study subjects, and the detection frequency varied with sex and age, being higher in females than in males (10.3% vs 8.0%) 166 and older subjects (P<0.01) ( data not shown).
Why don't they show the data?
I think the graphics should be integrated into the text, I think it does not exceed the number of pages.
Answer)
Authors represented the data as Supplementary Table S1 as recommended.
Thank you very much!

Reviewer 2 Report
Comments and Suggestions for Authors
The manuscript “Vitamin D status and reference interval measured by liquid chromatography-tandem mass spectrometry for the early adulthood to geriatric ages in South Korean population during 2017–2022.”
Is conceived to analyze 25-hydroxyvitamin D (25(OH)D) status of the South Korean population aged ≥20 years using 25(OH)D concentrations measured by liquid chromatography–tandem mass spectrometry.
The authors conducted a retrospective study to analyse the levels of this important analyte with the most advanced methodology, considered the gold standard.
I have a few major points to make:
- As they rightly pointed out themselves (line 263), such an ambitious study did not record data on the simultaneous intake of supplements, or fortified food. The retrospective nature of the analysis did not allow for such a clinical data collection procedure?
- The authors state that they analysed data from subjects undergoing a checkup, which included analysis of vitamin 25(OH)D; how could they analyse sera stored at 4°C? Does this mean that 25(OH)D analysis was performed immediately after collection? This fact should be made much clearer in the materials and methods, specifying that the analysis is conducted on the serum after centrifugation of the blood (insert method).
- The authors claim to perform an ANOVA with post-hoc comparison; which test did they use (Tukey, Bonferroni... etc?)
Minor points:
Line 92: Please explain centrifugation methods and how many hours after the blood collection analysis was performed.
Line 113: RIs stand for Reference Indices?
Line 127: please indicate which other variables were chosen for adjusting the multiple linear regression model.
Line 133: The sentence on the dissemination of results is not necessary for a research manuscript (or probably this sentence should be mentioned at the end, please let inform on manuscript preparation section).
Discussion:
Line 196: The authors referred to this study, but without any specification the readers cannot understand, thinking to a general population level already tested elsewhere. Please specify better.
Line 208: reference 22 did not cover the sentences about various disease connections with vit D, please add specific references.
Line 214: Korea stands for South Korea or all Korea (North+ South?)
L242: “Strategies for …. Are therefore considered.. “ Authors referred to some supplementation already done, or do they think that prevention should be considered?
L274: please add: “However, the deficiency was high..” because it is a confounding sentence.
Author Response
Major points
1.As they rightly pointed out themselves (line 263), such an ambitious study did not record data on the simultaneous intake of supplements, or fortified food. The retrospective nature of the analysis did not allow for such a clinical data collection procedure?
Answer)
Authors retrospectively collected data from data base of health checkups. We could not obtain information of intake of supplements, or fortified food. Authors described this limitation in Discussion section (Page 8, Line 262-263).
2.The authors state that they analysed data from subjects undergoing a checkup, which included analysis of vitamin 25(OH)D; how could they analyse sera stored at 4°C? Does this mean that 25(OH)D analysis was performed immediately after collection? This fact should be made much clearer in the materials and methods, specifying that the analysis is conducted on the serum after centrifugation of the blood (insert method).
Answer)
Authors analyzed data from subjects undergoing a health checkup including vitamin 25(OH)D test. Authors described this and pre-analytic procedures in the Materials and Methods section (Page 3, Line 91-95) as follows: The health checkups included the drawing of venous blood after an overnight fast to measure vitamin D. Venous blood were collected in serum tubes with clot activator. As per routine procedure, blood samples were centrifuged at 1,500 rpm for 10 minutes upon arrival in the lab and stored at 4 °C until they were analyzed within 7 days in the core laboratory.
3.The authors claim to perform an ANOVA with post-hoc comparison; which test did they use (Tukey, Bonferroni... etc?)
Answer)
Authors performed an ANOVA with Bonferroni’s post-hoc test. Authors described this in Materials and Methods section (Page 3, Line 120-121) as follows: blood collection were analyzed using one-way ANOVA with Bonferroni’s post-hoc comparison.
Minor points
1.Line 92: Please explain centrifugation methods and how many hours after the blood collection analysis was performed.
Answer)
Authors described the centrifugation method in Materials and Methods section (Page 3, Line 92-95) as follows: Venous blood were collected in serum tubes with clot activator. As per routine procedure, blood samples were centrifuged at 1,500 rpm for 10 minutes upon arrival in the lab and stored at 4 °C until they were analyzed within 7 days in the core laboratory.
2.Line 113: RIs stand for Reference Indices?
Answer)
RIs stand for reference intervals. Authors described this in Materials and Methods section (Page 3, Line 115) as follows:
2.3. Statistical analysis and calculation of reference intervals for serum 25(OH)D2, 25(OH)D3, and total 25(OH)D
3.Line 127: please indicate which other variables were chosen for adjusting the multiple linear regression model.
Answer)
Authors indicated variables for adjusting the multiple linear regression model in Materials and Methods section (Page 4, Line 128-130) as follows: Multiple linear regression analyses were performed to identify the factors associated with the total 25(OH)D concentration after adjusting for sex, age, month of blood collection, BMI, and hs-CRP.
4.Line 133: The sentence on the dissemination of results is not necessary for a research manuscript (or probably this sentence should be mentioned at the end, please let inform on manuscript preparation section).
Answer)
Authors deleted the sentence as recommended.
5.Line 196: The authors referred to this study, but without any specification the readers cannot understand, thinking to a general population level already tested elsewhere. Please specify better.
Answer)
Authors specified this in Discussion section (Page 7, Line 196-197) as follows: The mean total 25(OH)D concentration by LC-MS/MS method in the present study was 54.5 nmol/L for the South Korean general population,
6.Line 208: reference 22 did not cover the sentences about various disease connections with vit D, please add specific references.
Answer)
Authors added specified references as Reference 2-4 and 23 as recommended.
7.Line 214: Korea stands for South Korea or all Korea (North+ South?)
Answer)
Authors specified as South Korea (Page 7, Line 214).
8.L242: “Strategies for …. Are therefore considered.. “ Authors referred to some supplementation already done, or do they think that prevention should be considered?
Answer)
Authors described this in Discussion section (Page 8, Line 240-241) as follows: Strategies for vitamin D deficiency prevention should be considered for this subgroup.
9.L274: please add: “However, the deficiency was high..” because it is a confounding sentence.
Answer)
Authors added “the deficiency” in Line 272 as follows: However, the deficiency was high in young adults, especially in young females aged <30 years.
Thank you very much!
